# COVID-19-Related Testing, Knowledge and Behaviors among Severe and Chronic Non-Communicable Disease Patients in Neno District, Malawi: A Prospective Cohort Study

**DOI:** 10.3390/ijerph20105877

**Published:** 2023-05-19

**Authors:** Haules Robbins Zaniku, Moses Banda Aron, Kaylin Vrkljan, Kartik Tyagi, Myness Kasanda Ndambo, Gladys Mtalimanja Banda, Revelation Nyirongo, Isaac Mphande, Bright Mailosi, George Talama, Fabien Munyaneza, Emilia Connolly, Luckson Dullie, Dale A. Barnhart, Todd Ruderman

**Affiliations:** 1Neno District Health Office, Ministry of Health, Neno P.O. Box 52, Malawi; 2School of Global and Public Health, Kamuzu University of Health Sciences, Blantyre 312225, Malawi; 3Partners In Health/Abwenzi Pa za Umoyo (PIH/APZU), Neno P.O. Box 56, Malawi; 4Harvard College, Harvard University, Cambridge, MA 02138, USA; 5Gillings School of Global Public Health, University of North Carolina at Chapel Hill, Chapel Hill, NC 27599, USA; 6Malawi Epidemiology and Intervention Research Unit (MEIRU), Lilongwe P.O. Box 148, Malawi; 7Partners in Hope, Lilongwe P.O. Box 302, Malawi; 8Division of Pediatrics, College of Medicine, University of Cincinnati, 3230 Eden Ave., Cincinnati, OH 45267, USA; 9Division of Hospital Medicine, Cincinnati Children’s Hospital Medical Center, 3333 Burnet Ave., Cincinnati, OH 45529, USA; 10Partners in Health/Inshuti Mu Buzima (PIH/IMB), Kigali P.O. Box 3432, Rwanda; 11Department of Global Health and Social Medicine, Harvard Medical School, Boston, MA 02138, USA

**Keywords:** noncommunicable diseases, COVID-19 testing, COVID-19, Malawi, chronic disease

## Abstract

COVID-19-related knowledge and behaviors remain essential for controlling the spread of disease, especially among vulnerable patients with advanced, chronic diseases. We prospectively assessed changes over 11 months in COVID-19-related testing, knowledge, and behaviors among patients with non-communicable diseases in rural Malawi using four rounds of telephone interviews between November 2020 to October 2021. The most commonly reported COVID-19-related risks among patients included visiting health facilities (35–49%), attending mass gatherings (33–36%), and travelling outside the district (14–19%). Patients reporting having experienced COVID-like symptoms increased from 30% in December 2020 to 41% in October 2021. However, only 13% of patients had ever received a COVID-19 test by the end of the study period. Respondents answered 67–70% of the COVID-19 knowledge questions correctly, with no significant changes over time. Hand washing, wearing face masks and maintaining a safe distance were the most frequently reported strategies to prevent the spreading of COVID-19. Wearing face masks significantly improved over time (*p* < 0.001). Although the majority reported accurate knowledge about COVID-19 and enhanced adherence to infection prevention measures over time, patients commonly visited locations where they could be exposed to COVID-19. Government and other stakeholders should increase COVID-19 testing accessibility to primary and secondary facilities.

## 1. Introduction

Due to misconceptions and false information about COVID-19, assessing COVID-19 knowledge and behaviour regarding its transmission and acquisition methods is essential in controlling the outbreak [1]. A systematic review of 21 studies found that, on average, there has been good knowledge about COVID-19 among the general population [2]. However, previous studies in Malawi found that the general population had insufficient knowledge about COVID-19 and poor practices against its transmission [3,4]. Other studies have also found that knowledge and practice towards COVID-19 among patients with chronic diseases is poor [5,6,7].

People with pre-existing chronic diseases, including hypertension, diabetes mellitus, chronic renal failure, and ischemic heart disease, face an elevated risk of severe illness, hospitalization and death from COVID-19 [8,9,10]. COVID-19 vaccines effectively prevent severe illness [11,12], even among individuals with chronic diseases [8]. However, COVID-19 vaccination rates in low- and medium-income countries (LMICs) like Malawi remain low [13]. As of 11 March 2022, Malawi had 828,080 people (4.4% of its population) fully vaccinated against COVID-19 [14]. These low rates reflect a combination of inequitable distribution of global vaccines and vaccine hesitancy [15,16]. In this context of limited COVID-19 vaccination coverage, knowledge about COVID-19, access and acceptability of COVID-19 testing, and individual infection preventive measures continue to play a central role in mitigating the spread of COVID-19 [5,17]. These COVID-19 prevention measures, such as social distancing and mask-wearing, are especially critical for unvaccinated chronic disease patients because of their elevated risk of severe disease with COVID-19 infection [5]. However, while extensive literature exists on COVID-related knowledge and practices in the general population and among healthcare workers [18,19,20], fewer studies have focused on patients with non-communicable diseases (NCDs), especially in rural African areas [21]

In Malawi, surveys conducted in urban [3], primarily rural [4], and nationally representative populations [22] at the start of the COVID-19 pandemic (April–September 2020) found mixed evidence related to knowledge and practices surrounding COVID-19. For example, while many respondents understood that the virus could be spread through close contact with a COVID-19 case and through respiratory transmission [3,4], respondents also reported several misconceptions surrounding alternate transmission routes and COVID-19 severity, especially in rural settings [4,22]. Similarly, although almost all respondents reported taking some steps to prevent COVID-19, increased handwashing was the most commonly adopted preventive measure, with markedly fewer people practicing social distancing or masking, especially in rural areas settings [4,22]. However, we know these patterns of COVID-19-related knowledge and prevention could be different among patients with non-communicable diseases. These patients could have more opportunities to learn about COVID-19 due to frequent contact with the health system and be more highly motivated to adhere to COVID-19 prevention practices due to their elevated risk.

To better understand patients’ knowledge of COVID-19 prevention practices and vulnerability to COVID-19 infection, we conducted an open cohort study on COVID-19 knowledge, risks, symptoms and testing, and infection preventions practices among patients with complex NCD receiving care at clinics located at Neno District and Lisungwi Community Hospitals.

## 2. Materials and Methods

### 2.1. Study Design and Setting

We conducted a prospective open cohort telephone study among patients enrolled in NCD clinics at Neno District and Lisungwi Community Hospitals. These clinics provide the World Health Organization’s recommended Package of Essential Non-communicable Disease Interventions for first-referral hospitals (PEN-plus). The hospitals are located in Neno District, in the southern part of Malawi, with an estimated population of approximately 150,211 people [23], with 8758 people using electricity as their primary lighting source [24]. Partners In Health/Abwenzi Pa Za Umoyo (PIH/APZU), an international non-governmental organization, has partnered with the Ministry of Health (MOH) to strengthen health systems and improve health outcomes Neno district since 2007. In 2011, APZU/MOH established its first Chronic Care Clinic at Neno District Hospital, which provides longitudinal care for patients with an array of chronic diseases, including HIV and common NCDs, under one roof. In 2018, two PEN-Plus NCD clinics were established, which provide outpatient services for severe and complex NCDs such as sickle cell disease, rheumatic heart disease and type 1 diabetes [25,26]. PIH/APZU and MOH currently support care for over 4000 NCD patients, including approximately 500 patients with severe and chronic NCDs.

### 2.2. Study Population

All patients (a) enrolled in one of the PEN-Plus NCD clinics and (b) had a telephone number on their patient cards and were eligible to participate. Those who did not have severe or complex NCDs were excluded. For patients under 18 years of age or critically ill, their adult caregiver consented to participate in the study and responded on their behalf. At the start of the study, out of 450 patients enrolled in advanced NCD as of December 2020, 105 had personal phones. An additional 50 had received phones as part of a PIH/APZU initiative to maintain continuity of care during the COVID-19 pandemic, leading to an estimated telephone coverage of 34% for this study population (*n* = 155). Due to the small size of this clinical population, we invited all patients who met our inclusion criteria in each round to participate.

### 2.3. Data Collection

We repeated the quarterly telephone surveys to track changes in patients’ COVID-19-related information. The four rounds of data collection occurred in November–December 2020, March–April 2021, June–July 2021, and September–October 2021. In each round, we updated the list of eligible patients with phone numbers and made up to five attempts on five separate days to contact eligible participants by telephone. After five attempts, patients who could not be contacted were referred to the clinical team to verify patient well-being via an in-person visit. The telephone interviews lasted approximately 25–40 min.

The data collection tools were adapted from a questionnaire developed by the Partners in Health Cross-Site COVID-19 Cohort Research Network, a team of clinicians and researchers from eight PIH-supported countries and methodologists from Harvard Medical School and Partners In Health-Boston. The questionnaire included modules on patient and respondent demographics and patient health history. We collected data on patients’ COVID-19-related symptoms, COVID-19 risk factors and COVID-19 testing history, with caregivers responding on behalf of patients when necessary. We assessed the respondents’ COVID-related knowledge, which was assessed using questions adapted from Banda et al. [4] by asking respondents to report on their COVID-19-related knowledge, whether they were the patient or a caregiver. We used an open-ended question to understand what action each household was taking to prevent COVID-19 and data collectors were asked to code responses according to a pre-defined list of select-multiple options. In the fourth round of the survey, questions about COVID-19 knowledge were replaced with questions on COVID-19 vaccination which will be reported in another paper. The questions were translated to “Chichewa”, the local language programmed in CommCare, an open-source mobile application used to collect data. We recruited two enumerators to recruit study participants and administered each telephone survey interview using a CommCare application throughout the four data collection cycles. The enumerators received a one-day training on how to administer the questionnaire and use CommCare before the first round of data collection.

### 2.4. Data Analysis

We described the population of respondents using frequencies and percentages for categorical variables and medians and interquartile ranges (IQR) for continuous variables. For each round, we reported the proportion of patients who experienced COVID-19-related risk factors, including visiting the health facility, attending a mass gathering, and travelling outside Neno over the past two weeks. We reported the proportion of patients ever reporting COVID-like symptoms, defined as either loss of taste or smell or cough and at least one of fever, chills, muscle aches and shortness of breath. Similarly, we reported the proportion of patients who ever experienced a COVID-19 test. We assessed respondents’ COVID-19-related knowledge by assessing whether they agreed or disagreed with a series of six statements and reported the percentage of respondents in each round that answered a specific question correctly, with responses of “don’t know” classified as incorrect. We also calculated an overall summary of COVID-19 knowledge by dividing the number of questions answered correctly by the number of questions attempted by the respondent. We assessed the proportion of respondents using the radio, Ministry of Health, friends and/or family, television, and social media as a source of knowledge on COVID-19. Finally, we reported the percentage of households engaging in various infection prevention control strategies. We visualized our data using bar charts to report population averages for each outcome at each wave. To test whether there were significant changes over time in overall COVID-19 knowledge, which was a continuous measure with a theoretical range of 0% (no correct responses) to 100% (all six responses were correct), we used a linear regression model with an indicator variable for wave two and wave 3 to test the null hypothesis neither the coefficient for wave two nor the coefficient for wave three were significantly different than zero. This null hypothesis would correspond to no significant differences across the waves. Our linear regression model accounted for within-patient clustering over time due to repeated responses from the same patient across multiple time points. The remaining outcomes in the study were binary. For these outcomes, we tested for significant changes over time using logistic regression outcomes that similarly included indicator variables for each study wave and accounted for within-patient clustering over time. Data were analyzed in Stata version 15.1.

## 3. Results

### 3.1. Sociodemographic Characteristics of Patient and Respondents

Over four rounds of data collection, we enrolled 192 participants, most of whom were enrolled in the first round (*n* = 145, 75.5%) or the third round (*n* = 42, 21.9%). The sample sizes for the four rounds were 145, 131, 155, and 126, respectively. Of the 192 patients in the study, 101 (52.6%) were female; their median age was 50 years (IQR: 32, 68), with over one-third aged 60 years or older. Type 1 diabetes (*n* = 50, 26.0%), hypertension (*n* = 34, 17.7%), and chronic heart failure (*n* = 38, 19.8%) were the most commonly reported conditions. Of most patients, 114 (59.4%) had multiple conditions. Out of the 192 respondents, 126 (65.6%) were the patients themselves, the median age was 47 (IQR: 35, 62), and 111 (57.8%) were female (Table 1).

### 3.2. COVID-19 Risks, Symptoms and Testing among Patients

In all four rounds of data collection, the most commonly reported COVID-19-related risks experienced by patients in the past two weeks were visiting the health facility (range: 35–49%), followed by attending mass gatherings (range: 33–36%) and travelling outside Neno (range: 14–19%). Visiting traditional healers, travelling outside Malawi, or being in contact with a COVID-19 case were rarely reported. Only the proportion of patients who reported visiting health facilities changed significantly over time (*p* < 0.0183). In the first round, one-third (30%) of participants reported having ever experienced COVID-like symptoms, and this increased to 41% in the fourth round (*p* < 0.0132). Only 13% of the participants in the fourth round reported ever receiving a COVID-19 test at any point in the study, and only one patient in the fourth round reported testing positive. Statistically, we found no significant change over time regarding receipt of COVID-19 testing (Figure 1).

### 3.3. COVID-19 Knowledge among Respondents

Respondents answered about two-thirds of the COVID-19 knowledge questions correctly, with no significant changes over time (67–70%). Overall, COVID-19 knowledge questions where “Agree” was the correct answer had a much higher percentage of respondents (range: 78–97%) answering correctly when compared to questions where “Disagree” was the correct answer (range: 22–31%) (Figure 2).

Respondents most commonly cited the radio (range: 81–85%), Malawi Ministry of Health (range: 79–86%), and friends or family (range: 43–59%) as sources of COVID-19 knowledge, while television (range: 7–8%) and social media (range: 3–5%) were the least common sources of knowledge (Figure 3). In addition, respondents citing family or friends as a source of COVID-19 knowledge reduced significantly over time (*p* < 0.0093).

### 3.4. Infection Prevention Strategy among Households

The most commonly practised infection prevention strategies were washing hands (range: 90–95%), wearing facemasks (range: 63–96%) and maintaining a safe distance (range: 54–68%). The adoption of wearing facemasks and avoiding crowded areas increased over time (*p* < 0.001). Hygienic coughing and sneezing significantly decreased over time (*p* < 0.0072). In all four rounds, less than a quarter of households reported avoiding going out, using hand sanitizer, and avoiding within-country travel, with no significant changes over time (Figure 4). The use of infection prevention strategies classified as “other” dropped significantly from 17% to 5% over time (*p* < 0.0013).

## 4. Discussion

In our prospective telephone-based open cohort study of COVID-19 patients with severe and chronic NCDs in rural Malawi, we found frequent exposure to COVID-19 risks with low COVID-19 testing. The participants had moderate knowledge about COVID-19, which did not improve over time, but high engagement in COVID-19 prevention activities, which did improve over time. Our study found visiting health facilities was the most common risk factor, likely reflecting high patient engagement in NCD care. However, visiting health facilities dropped sharply during rounds two (March–April 2021) and four (September–October 2021). This reduction is likely due to reduced health service utilization during large waves of COVID-19 between early January and late March 2021 and late June and September 2021 [27]. Similar health service utilisation decreases during the waves have been reported in 20 countries [28]. Our findings suggest that our patients were skipping appointments to reduce their risk of COVID-19 at the expense of receiving appropriate care and treatment for their NCDs.

By the fourth round of our study, 41% of patients reported having experienced COVID-like symptoms, but only 13% had ever received COVID-19 testing. Although this level of testing is sub-optimal, testing among our cohort in December 2020 (8%) was higher than what was reported by Banda et al. in a survey conducted among current and former residents of Karonga District, Malawi and their siblings in November 2020 (5.9%) [29]. Inadequate access to testing at health facilities is likely the primary cause of this low coverage. Malawi relied exclusively on PCR testing to diagnose COVID-19 from the pandemic's start in March 2020 until rapid diagnostic tests (RDTs) received government approval in January 2021. However, even after RDTs became available at Neno District Hospital, supplies were limited, and RDTs were only used for symptomatic patients or oxygen saturation <92% [30].

Furthermore, since the RDTs were only available at the hospital and not at more local primary-level health facilities, patients with mild symptoms may have chosen not to get tested due to the distance to the facilities. Only one patient in our study reported testing positive for COVID-19. Given the low accessibility of testing and high COVID-related stigma [31], it is likely that this finding reflects widespread under-testing and under-reporting rather than low COVID-19 prevalence. Seroprevalence studies done in Malawi and other countries have reported increasing seroprevalence for SARS-CoV-2 over time (12.3%, 14.3%, 51.5%) [32,33]. These findings underscore the importance of increasing the accessibility of decentralized COVID-19 testing, especially for vulnerable populations such as chronic disease patients and their household members.

Knowledge surrounding COVID-19 was moderate but did not improve over time. Our findings were very similar to Banda et al. [29], who found that approximately 90% of rural respondents understood the virus was transmitted through respiratory droplets, approximately 70% knew that there was no effective treatment for COVID-19, but only approximately 75% knew that asymptomatic transmission was possible. However, our patients generally had better knowledge than urban respondents from Li et al. [3]. Because Li et al. conducted household surveys while our research and Banda’s research used telephone-based data collection, these differences could reflect disparities in access to information among households that do not own a telephone. Since COVID-19 was recognized as a health risk in 2019, the Ministry of Health in Neno and it is implementing partners have been disseminating information related to COVID-19 transmission, prevention, and access to testing through a public address van, radio, and health talks. However, the lack of improvements over time, especially surrounding persistent beliefs in misconceptions around COVID-19, suggests that additional efforts are needed to address and correct existing misinformation.

In general, household engagement in infection prevention strategies was extremely high. Increased handwashing was a widespread response early in the pandemic, as has been reported previously in Malawi [3,29]. We also observed statistically significant improvements over time for household use of two key infection prevention measures: wearing face masks and avoiding crowded places. Maintaining safe distances also increased over time, although this difference was insignificant. Throughout the COVID-19 pandemic, the government of Malawi has been implementing mandatory mask strategies when visiting health facilities, markets, and political gatherings. Previous research in Malawi has observed similar improvements in using facemasks and social distance over time [29]. However, as evidenced by the fact that a third of patients attended a festival or mass gathering in the past two weeks across all four rounds of data collection, adherence to these infection prevention strategies is not perfect.

Our study has several limitations. Firstly, our sample size was small. However, we invited all patients who met our inclusion criteria in each round to participate to mitigate this. Secondly, this telephone survey was conducted among severe and chronic NCD patients in rural Malawi and is not generalizable to urban areas, other rural populations, or individuals without access to cellular service. However, because the clinical team distributed phones to low-income NCD patients to support continuity of care during COVID-19, our study population may better capture low-income patients generally without mobile phones than other comparable telephone surveys. Thirdly, using a phone survey could have led to selection bias. However, additional 50 mobile phones were distributed to those who did not have mobile phones. Fourthly, because our target population were those with severe and chronic non-communicable diseases, our sample had a large proportion of type-1 DM, which may lead to selection bias and could not represent the target population.

Furthermore, all data were self-reported, which can be prone to misreporting. In particular, reporting COVID-19-compatible symptoms or testing could be prone to recall errors and may not be very specific or sensitive for diagnosing COVID-19 cases. However, we do believe that our definition of COVID-19 compatible symptoms is sufficient to identify individuals who could benefit from additional COVID-19 testing and is a useful measure for understanding the current gaps in testing coverage. Similarly, social desirability bias could have impacted our study, especially since hospital staff could be perceived as authority figures by patients. This bias might have affected reporting for COVID knowledge questions if patients were afraid to disagree with false statements. However, we tried to minimize this by recruiting enumerators who were not medical providers. Lastly, our sample size was small, leading to imprecision in our estimates.

Despite these limitations, our study provides valuable information on COVID-19-related testing, knowledge, and behaviors among highly vulnerable population patients with severe chronic NCDs in rural Malawi. Until COVID-19 vaccines are globally available in high- and low-income countries, these patients will remain vulnerable to COVID-19 infection and severe illness. Our research suggests several concrete steps are still needed to support these patients. First, there is a persistent need for decentralization and increased capacity for testing in rural Malawi. Secondly, the government and other stakeholders should consider providing social support for patients to follow COVID prevention measures. These should include, at a minimum, high-quality masks that can be used by patients with NCDs and their caregivers when they travel to health facilities or other crowded areas. Finally, there is a continued need for advocacy for increased availability of COVID-19 vaccines and educational interventions to encourage vaccine uptake among NCD patients and their household members.

## 5. Conclusions

Participants had moderate knowledge about COVID-19 and did not improve over time. Patients had frequent exposure to COVID-19 risks with low COVID-19 testing. However, there was a high engagement in COVID-19 prevention activities, which did improve over time. Visiting health facilities was the most common risk factor, likely reflecting high patient engagement in NCD care.

## Figures and Tables

**Figure 1 ijerph-20-05877-f001:**
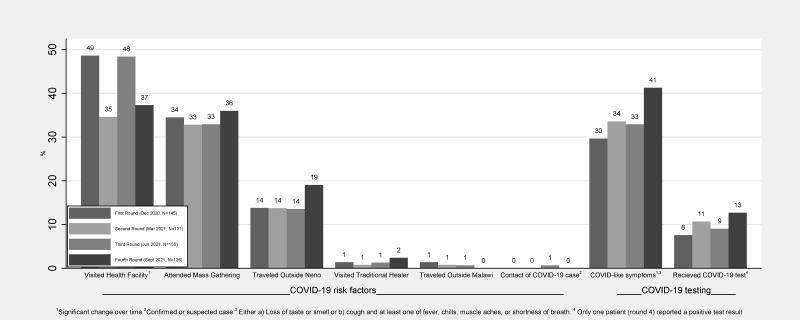
COVID-19 risks, symptoms, and testing among patients.

**Figure 2 ijerph-20-05877-f002:**
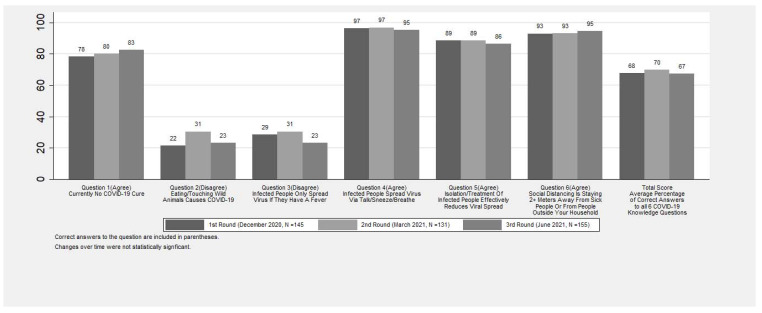
Percent of respondents who answered COVID-19 questions correctly.

**Figure 3 ijerph-20-05877-f003:**
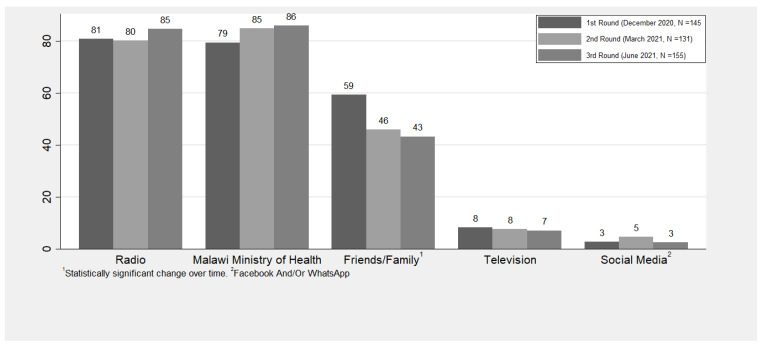
Percent of respondents utilizing each source of knowledge.

**Figure 4 ijerph-20-05877-f004:**
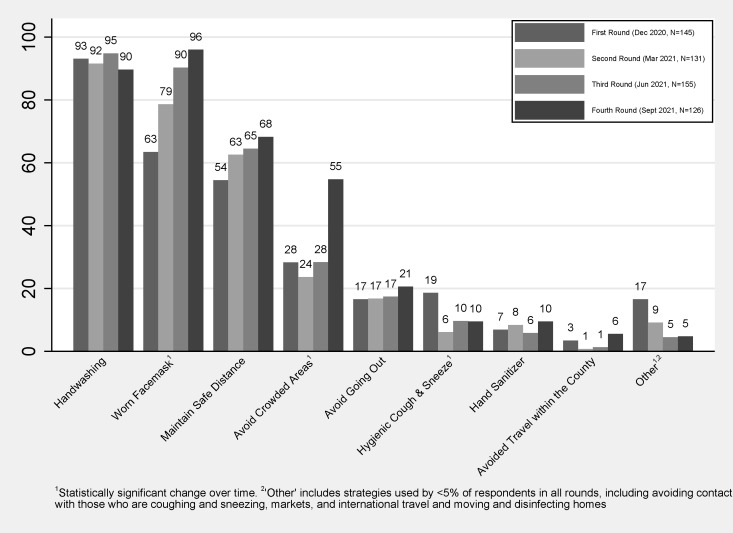
Percent of households engaging in infection prevention strategies.

**Table 1 ijerph-20-05877-t001:** Sociodemographic characteristics of patient and respondent (N = 192).

	N (%)
Round of Enrollment	
First Round (December 2020)	145 (75.5%)
Second Round (March 2021)	5 (2.6%)
Third Round (June 2021)	42 (21.9%)
Fourth Round (September 2021)	0 (0.0%)
Patient Sex	
Female	101 (52.6%)
Male	91 (47.4%)
Patient Age	
≤5 years	9 (4.7%)
6–17	16 (8.3%)
18–39	39 (20.3%)
40–59	57 (29.7%)
≥60	71 (37.0%)
Patient’s Clinical Program	
Type 1 Diabetes	50 (26.0%)
Hypertension	38 (19.8%)
Chronic Heart Failure	34 (17.7%)
Type 2 Diabetes	20 (10.4%)
Chronic liver disease/cirrhosis	16 (8.3%)
Rheumatic/another heart disease	17 (8.9%)
Chronic Kidney Disease	10 (5.2%)
Other	7 (3.6%)
Patient’s number of conditions	
1	78 (40.6%)
2	70 (36.5%)
≥3	44 (22.9%)
Respondent type	
Patient	126 (65.6%)
Parent	20 (10.4%)
Child	23 (12.0%)
Spouse	9 (4.7%)
Grandparent	6 (3.1%)
Other	8 (4.2%)
Respondent’s Sex	
Female	111 (57.8%)
Male	81 (42.2%)
Respondent’s Age	
18–39	65 (33.9%)
40–59	73 (38.0%)
≥60	54 (28.1%)

## Data Availability

The data presented in this study are available on request from the corresponding author. The data are not publicly available due to privacy and confidentiality.

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
