# Peer review of "COVID-19-Related Testing, Knowledge and Behaviors among Severe and Chronic Non-Communicable Disease Patients in Neno District, Malawi: A Prospective Cohort Study"

_ijerph, 2023, doi:10.3390/ijerph20105877_

Round 1
Reviewer 1 Report
Thank you very much for inviting me to review the paper entitled “COVID-19-related testing, knowledge and behaviors among severe and chronic non-communicable disease patients in Neno District, Malawi: A prospective cohort study.” Its a on time research topic about COVID-19-related knowledge and behaviors. This research tries to know changes over 11 months in COVID-19-related testing, knowledge, and behaviors among 31 patients with non-communicable diseases in rural Malawi using four rounds of telephone interviews between November 2020 to October 2021. Overall, I found it an interesting research topic. I hope my comments help you to improve the paper and make it publishable.
The data has been collected for an interesting context, Malawi, a location which we don’t know too much in the academia. Most importantly, the authors clearly defined the research objectives of the at the beginning of the paper. and the way of writing is simple and easily understandable. I enjoy reading the paper.
My recommendation to the authors is adding one paragraph at the beginning of INTRODUCTION section and discussing in more details the importance of COVID-19-related knowledge and behaviors. Before jumping to context of study and what going on in Malawi during Covid-19 pandemic, we need to know a little bit more about the research topics and its importance in the academia, what we know from previous research? What knowledge or understanding is missing? Later you can develop your research contributions on base of this research gap.
I wish you success in your research
Reviewer 2 Report
Thank you for allowing me to review the manuscript. The study aimed to assess whether knowledge of COVID-19 in rural areas of Malawi improved over time. I am afraid that this study has limited use in public health related research and informing public health policies. Please also find below other issues.
- Data analysis: please specify what statistical tests were used to compare the proportions between groups. I am concerned that the groups were not independent but also not entirely dependent because some but not all participants were involved in any two rounds of survey. Coupled with the size of the sample, the results are not reliable.
- I doubt if it is meaningful to ask the same participate about the knowledge in COVID-19 over time. If one knew that the virus was transmitted via respiratory droplets, he/she would likely give the same answer in the subsequent rounds of the survey. It is only meaningful if the sample size is very large that allows us to measure how much the general public knows about COVID-19 over time.
- It is surprising that there were more participants with type 1 DM than type 2. If the target population of your study is those who had chronic diseases. I am afraid it is not a representative sample.
- As mentioned in the text that 8758 in 150211 people used electricity as their main source of lightning, did using a phone survey lead to selection bias that those who had a high socioeconomic status had a high chance to have access to telephones?
Reviewer 3 Report
The authors chose a very current topic and carried out extensive research. First of all, congratulations on your research!
In my opinion, the work contributes greatly to the evaluation of some consequences of the pandemic crisis and to the identification of different forms of behavior.
Although I do not know the rules applied in Malawi, the authors analyzed the available data in an understandable and thoughtful manner.
It would have enriched the study if the authors had briefly indicated at the beginning whether similar research is or has been conducted on the topic.
The methodology is excellently structured, the language is easy to understand.
On a mental level, the thought arc of the study is easy to understand and consistent.
The authors drew relevant conclusions from the data using the appropriate methodology.
I recommend for consideration the refinement of the wording "Second, we urge the government" in point 4.
Reviewer 4 Report
The article was well written by using prospective assesment of awareness in regards to COVID-19 management. As we evolved over the pandemic and gone through different strains of infection thr article outlined how public got awareness through numerous government initiatives and healthcare workers extreme hard work during those testing times. I would also like to see how these healthcare awarness steps changed the behaviours of the patients and the authors to mention more in depth/ detail in this regards like how they handled self isolations, social distancing, awarness about when to get tested, vaccination etc.' Overall well written and organized article focusing on Malawi public health.
Round 2
